# Association of CYP2C9*2 Allele with Sulphonylurea-Induced Hypoglycaemia in Type 2 Diabetes Mellitus Patients: A Pharmacogenetic Study in Pakistani Pashtun Population

**DOI:** 10.3390/biomedicines11082282

**Published:** 2023-08-16

**Authors:** Asif Jan, Muhammad Saeed, Ramzi A. Mothana, Tahir Muhammad, Naveed Rahman, Abdullah R. Alanzi, Rani Akbar

**Affiliations:** 1Department of Pharmacy, University of Peshawar, Peshawar 25000, Pakistan; naveedrahman3216@gmail.com; 2District Headquarter Hospital (DHQH) Charsadda, Charsadda 24430, Pakistan; 3Department of Pharmacy, Qurtaba University of Science and Technology, Peshawar 25000, Pakistan; saeedrph@qurtuba.edu.pk; 4Department of Pharmacognosy, College of Pharmacy, King Saud University, Riyadh 11451, Saudi Arabia; rmothana@ksu.edu.sa (R.A.M.); aralonazi@ksu.edu.sa (A.R.A.); 5Molecular Neuropsychiatry & Development (MiND) Lab, Campbell Family Mental Health Research Institute, Centre for Addiction and Mental Health, Toronto, ON M6J 1H4, Canada; t.muhammad@mail.utoronto.ca; 6Institute of Medical Science, University of Toronto, Toronto, ON M5S 1A8, Canada; 7Department of Pharmacy, Abdul Wali Khan University, Mardan 23200, Pakistan; syedashah61@yahoo.com

**Keywords:** cytochrome P450 (CYP450), pharmacogenetics, CYP2C9*2 allele, sulphonylurea, hypoglycaemia, Pashtun, Pakistan

## Abstract

Polymorphism in cytochrome P450 (CYP) 2C9 enzyme is known to cause significant inter-individual differences in drug response and occurrence of adverse drug reactions. Different alleles of the *CYP2C9* gene have been identified, but the notable alleles responsible for reduced enzyme activity are *CYP2C9*2* and *CYP2C9*3*. No pharmacogenetic data are available on *CYP2C9*2* and *CYP2C9*3* alleles in the Pakistani population. In Pakistan, pharmacogenetics, which examines the relationship between genetic factors and drug response, are in the early stages of development. We, for the first time, investigated the association between the *CYP2C9* variant alleles *CYP2C9*2* and *CYP2C9*3* and the incidence of hypoglycaemia in patients with Type 2 diabetes mellitus (T2DM) receiving sulphonylurea medications. A total of *n* = 400 individuals of Pashtun ethnicity were recruited from 10 different districts of Khyber Pakhtunkhwa, Pakistan to participate in the study. The study participants were divided into two distinct groups: the case group (*n* = 200) and the control group (*n* = 200). The case group consisted of individuals with T2DM who were receiving sulphonylurea medications and experienced hypoglycaemia with it, whereas the control group included individuals with T2DM who were receiving sulphonylurea medication but did not experience sulphonylurea-induced hypoglycaemia (SIH). Blood samples were obtained from study participants following informed consent. DNA was isolated from whole blood samples using a Wiz-Prep DNA extraction kit. Following DNA isolation, *CYP2C9* alleles were genotyped using MassARRAY sequencing platform at the Centre of Genomics at the Rehman Medical Institute (RMI). The frequency of *CYP2C9*2* (low-activity allele) was more frequent in the diabetic patients with SIH compared to the control group (17.5% vs. 6.0%, *p* = 0.021). The frequency of its corresponding genotype *CYP2C9*1/*2* was higher in cases compared to the control group (10% vs. 6% with *p* = 0.036); the same was true for genotype *CYP2C9*2/*2* (7% vs. 3.5% with *p* = 0.028). Logistic regression analysis evidenced potential association of *CYP2C9*2* allele and its genotypes with SIH. When adjusted for confounding factors such as age, weight, sex, mean daily dose of sulphonylurea, and triglyceride level, the association between the *CYP2C9*2* allele and hypoglycaemia remained consistent. Confounding factors played no role in SIH (insignificant *p*-value) because both groups (cases and controls) were closely matched in term of age, weight, sex, mean daily dose of sulphonylurea, and triglyceride levels. Our study suggests that genetic information about a patient’s *CYP2C9* gene/enzyme can potentially assist physicians in prescribing the most suitable and safest drug, based on their genetic make-up.

## 1. Introduction

Type 2 diabetes mellitus (T2DM) is considered a 21st century epidemic [1]. It affects approximately 537 million people worldwide, including 19.4 million in Pakistan [2]. Sulphonylureas are a class of oral anti-diabetic medication frequently used in the management of T2DM [3]. The mechanism of action of sulphonylureas involves stimulating the release of insulin from pancreatic beta cells, which leads to a reduction in blood glucose level [4]. However, sulphonylurea drugs secrete insulin independently of blood glucose level; as a consequence, patients with T2DM undergoing sulphonylurea therapy are at high risk of developing hypoglycaemia (lower blood sugar level than normal), which has potentially severe consequences [5]. Hypoglycaemia poses a great risk and can be more dangerous and life-threatening for individuals with T2DM and concurrent cardiovascular complications [6]. Known risk factors for sulphonylurea-induced hypoglycaemia (SIH) include low haemoglobin (Hb)A1c, old age, long duration of diabetes, comorbid conditions (such as congestive heart failure, coronary artery disease, ischemic heart disease, and renal insufficiency), taking multiple medications (polypharmacy), use of long-acting sulphonylurea preparations, and pharmacogenetic factors [7,8]. Genetic polymorphisms in the enzymes responsible for metabolizing sulphonylurea affect pharmacokinetics and pharmacodynamics of these drugs, leading to altered drug metabolism, clearance rate, and drug action. Individual genetic makeup is a critical factor in drug selection and determination of the appropriate drug dose. Some individuals metabolize the drug slowly, resulting in higher levels of the drug in their bloodstream over time, which can lead to a prolonged hypoglycaemic action. On the other hand, individuals who metabolize sulphonylureas rapidly experience lower drug levels, potentially leading to reduced efficacy and a higher risk of treatment failure [9,10].

Cytochrome P450 (*CYP450*) enzymes play a vital role in drug metabolism. These *CYP450* enzymes are primarily present in the liver, but are also present in other tissues, including the pancreas, kidneys, and gastrointestinal tract [11]. The Human Genome Project has identified at total of *n* = 57 human *CYP450* enzymes; a significant portion of drug metabolism is carried out by a subset of these enzymes. Approximately 90% of drugs are metabolized by six major CYP enzymes, namely, *CYP1A2, CYP2C9, CYP2C19, CYP2D6, CYP3A4,* and *CYP3A5* [12]. The commonly used second-generation sulphonylureas, such as glibenclamide, glipizide, and glimepiride, are primarily metabolized by the cytochrome P450 (CYP) 2C9 enzyme [13]. Two well-known genetic variants of the *CYP2C9* gene are *CYP2C9*2* (Arg144Cys) and *CYP2C9*3* (Ile359Leu). These variants lead to amino acid substitutions in the *CYP2C9* enzyme, resulting in lower/decreased enzyme activity [14,15]. Different phenotypes (normal metabolizers, intermediate metabolizers, and poor metabolizers) exist, based on the genotypes of CYP alleles [16]. Patients who carry two identical copies of the wild-type CYP alleles (*CYP2C19*1/*1*) are classified as normal metabolizers. In this case, the metabolizing ability of enzymes is optimal. Heterozygous genotypes, where one copy of the allele is wild-type and the other copy is a reduced-function or loss-of-function allele (**1/*2* and **1/*3*), or homozygous genotypes, with two copies of reduced-function alleles (**2/*2*), result in an intermediate-metabolizer phenotype. In this case, individuals exhibit reduced enzyme activity compared to normal metabolizers. Homozygous genotypes with two copies of loss-of-function alleles (**3/*3*) or heterozygous genotypes with one copy of reduced-function and another of loss-of-function alleles (**2/*3*) are classified as poor metabolizers. These individuals have significantly impaired or absent enzyme activity, resulting in a significant impact on drug metabolism [16,17].

Genetic information about a patient’s *CYP2C9* enzymes helps physicians to prescribe the most suitable and safest drugs based on individual genetic make-up. The *CYP2C9* enzyme is responsible for metabolizing approximately 13% of clinically available drugs [18]. In Pakistan, where over 2.6 billion unit doses of drugs are dispensed annually, it is estimated that more than 332 million unit doses are metabolized by the *CYP2C9* enzyme. According to the research study of Ahmed et al., around 20% of Pakistan’s population carries a *CYP2C9* genotype that contains at least one low-activity allele. This indicates that over 66 million doses of drugs dispensed annually in Pakistan may not have the desired effects for patients with a low-activity *CYP2C9* allele. For drugs that require activation through the *CYP2C9* enzyme, patients with a low-activity allele may experience a lack of response. Conversely, if a drug is inactivated by *CYP2C9*, increased frequency and severity of adverse effects could be expected in individuals with a low-activity allele [19]. In Pakistan, pharmacogenomic data about cytochrome P450 enzymes are very limited. The frequencies of *CYP2C9*2* and *CYP2C9*3*, which are responsible for the low activity of the enzyme, are not known in the Pakistani cohort. The aim of the present case–control study was to screen patients with T2DM from the Pakistani Pashtun population for low-activity enzyme alleles and assess their possible role in SIH. Our study is an important step towards understanding the genetic factors influencing drug response in the studied population.

## 2. Material and Methods

### 2.1. Case and Control Definition

The cases were defined as individuals with T2DM who have experienced hypoglycaemia while being treated with sulphonylurea medications. Hypoglycaemia refers to a condition where blood sugar levels drop below normal. On the other hand, the controls were individuals with T2DM who were receiving treatment with sulphonylurea medications but have not experienced hypoglycaemia. The cases and controls were differentiated by their response to the sulphonylurea treatment in terms of experiencing hypoglycaemia.

### 2.2. Subject Selection

A total of 400 individuals (T2DM/controls *n* = 200 and T2DM patients with SIH/cases *n* = 200) of Pashtun ethnicity, belonging to 10 different districts (Peshawar, charsadda, mardan, Kohat, Noweshera, swabi, Bannu, Karak, Dir and Swat) of Khyber Pakhtunkhwa, Pakistan, were included in this study. Patients with and without SIH were registered at the endocrinology units of three tertiary care hospitals: Lady Reading Hospital (LRH) Peshawar, Hayatabad Medical Complex (HMC) Peshawar, and Khyber Teaching Hospital (KTH) Peshawar. These hospitals provided specialized care for patients with endocrine disorders, including type 2 diabetes mellitus (T2DM). To ensure accurate comparisons, the cases (T2DM patients with SIH) were matched with the control group (T2DM) in terms of age, gender, and ethnicity. Written informed consent was obtained from all the participants to ensure their voluntary participation in the study. In the case of illiterate or uneducated patients, the informed consent form was read and explained to them in the local Pashtu language. If the patient agreed to participate, a relative or attendant signed the consent form on their behalf. Detailed demographic information and clinical parameters of the patients were collected using a carefully designed proforma. These parameters included relevant medical history, current medications, renal function/glomerular filtration rate, BMI, triglyceride levels, HbA1c level, and other factors that could potentially influence the development of SIH. Inclusion criteria for cases were as follows: (i) T2DM with SIH; (ii) Patient of age in the range of 30 to 80 years, and (iii) Patient from the Pashtun population. Individuals with mental disorders, age below 30 years, and those presenting chronic infections such as HCV, HBV, or malignancies were excluded from the study.

### 2.3. Ethical Approval

The study acquired ethical approval from the ethical committee of the Department of Pharmacy, University of Peshawar, with approval number 907/PHAR. All procedures conducted during the study adhered to the principles outlined in the Helsinki Declaration of 1975.

### 2.4. Collection of Blood Samples

Blood samples were collected from the study individuals by a trained nurse using aseptic procedures. The blood was drawn from the median cubital vein, typically located in the inner elbow area. Three millilitres of whole blood were collected from each participant using EDTA tubes, which were properly labelled to ensure accurate identification and traceability of the samples. After collection, the EDTA tubes containing the blood samples were stored at a temperature of −10 °C to preserve the integrity of the samples until further analysis or testing.

### 2.5. DNA Extraction and Quantification

Deoxyribonucleic acid (DNA) was extracted from 200 microlitres (μL) of whole blood samples obtained from cases and controls. The Wiz-Prep DNA extraction kit (Wiz-Prep no. W54100) was used for the extraction process, following the guidelines provided by the manufacturer with the kit. After successful DNA extraction, the quantification of the extracted DNA was conducted using the Invitrogen Qubit™3, a fluorometer-based system designed specifically for DNA quantification. The final DNA concentration was adjusted to 5 ng/μL.

### 2.6. SNPs Selection and Genotyping

Well-known genetic variants of the *CYP2C9* gene, namely, *CYP2C9*2* (Arg144Cys) and *CYP2C9*3* (Ile359Leu), were selected to assess its role in SIH. These variants are known to cause amino acid substitutions in the *CYP2C9* enzyme, leading to decreased enzyme activity. The genotyping of the selected variants was performed using the Sequenom MassARRAY genotyping platform at the Centre of Genomics, Rehman Medical Institute (RMI), Peshawar, via collaboration. The Sequenom MassARRAY genotyping method is a high-throughput genotyping technology that utilizes matrix-assisted laser desorption/ionization time-of-flight mass spectrometry (MALDI-TOF MS) to accurately and efficiently genotype genetic variations.

### 2.7. Statistical Analysis

The statistical analysis was carried out using IBM SPSS (Statistical Package for Social Sciences) version 24. The key variables selected for analysis included gender, advanced age, BMI, concurrent medications, triglyceride levels, geographical area (districts), smoking status, lifestyle, exercise, diet, occupation, and the selected genetic variants in *CYP2C9*. To assess the genetic variants conformity with the Hardy–Weinberg equilibrium (HWE), a chi-square (χ^2^) test was performed. The HWE test evaluated whether the observed genotype frequencies in the population differed significantly from the expected frequencies. To determine the difference in the distribution of allelic and genotypic frequencies between the cases (T2DM patients with SIH) and the control (T2DM patients), the χ^2^ test was used. To examine the association between the selected genetic variants in the *CYP2C9* gene with hypoglycaemia, a binary logistic regression test was performed. Logistic regression is a statistical technique used to analyse the relationship between independent variables (i.e., genetic variants) and a binary dependent variable (i.e., hypoglycaemia). A probability value (*p*-value) of less than 0.05 was considered statistically significant.

## 3. Results

### 3.1. Study Subjects’ Characteristics

Detail socio-demographics, biochemical features, and comorbidities prevalence in study subjects (cases and controls) are given in Table 1, Table 2 and Table 3. A total of *n* = 400 confirmed T2DM patients (*n* = 200 patients with SIH and *n* = 200 without SIH), of age in the range of 30 to 80 years, were included in this study. No significance differences (*p* > 0.05) in mean age and weight of cases vs. controls were observed. Among the study participants, 84% were male and 16% were female. Moreover, 76% of the participants were married, whereas 24% were unmarried. Additionally, 46% of participants were cigarette smokers, while 29% of the study participants were non-smokers. The use of Naswar (a local smokeless tobacco product) was reported to be high among study participants. It is worth noting that a significant proportion of the study subjects were illiterate and came from lower socio-economic backgrounds. Additionally, almost all patients (95%) we reported to have a positive family history of T2DM. Comorbidities prevalence were slightly more frequent in T2DM patients with SIH compared to the control participants, but the difference was not statistically significant (*p* > 0.05; details given in Table 2). The two groups showed no notable variations in sulphonylurea mean daily dose (*p* = 0.998 for glimepiride and *p* = 0.761 for gliclazide), urea level (*p* = 0.213), creatinine level (*p* = 0.982), and HbA1C (*p* = 0.991)—all details are listed in Table 3.

### 3.2. Allelic Frequencies

The allelic frequencies of *CYP2C9*1, CYP2C9*2,* and *CYP2C9*3* in the cases and controls are listed in Table 4. The notation *CYP2C9*1* refers to the wild-type allele; *1 allele represents the reference sequence of the gene. In the studied population, we reported that 84.0% of cases and 91.5% of controls carried the *CYP2C9*1*/wild-type allele. The *CYP2C9*2* low-activity allele was more frequent in the diabetic patients with SIH compared to the control group (17.5% vs. 6.0%). On the other hand, the *CYP2C9*3* (loss of enzyme activity) allele was present in seven (3.5%) diabetic patients with SIH and in five (2.5%) individuals belonging to control group.

### 3.3. Genotype Frequencies

The prevalence of *CYP2C9*1/*1, *1/*2, *2/*2, *2/*3,* and **3/*3* genotypes are listed in Table 5. The frequency of *CYP2C9*1/*1* (reference genotype) between the cases and controls was (79% vs. 89%), whereas the frequency of genotype **1/*2* was higher in the cases compared to the control group (10% vs. 6%, with *p* = 0.036); the same was true for genotype **2/*2* (7% vs. 3.5%, with *p* = 0.028). In contrast, the genotype **1/*3* was underrepresented in the cases compared to the control group (1% vs. 1.5%), albeit with *p*-value of 0.344. Genotypes **2/*3* and **3/*3* were not reported, either in the cases or controls.

### 3.4. Risk of Hypoglycaemia in CYP2C9 Carriers

Our study reported higher carriage rate of low-activity allele *CYP2C9*2* in the case group compared to the control (OR = 0.102, 95% confidence interval (CI): 0.08–3.08, *p* = 0.021, Table 4). The frequency of *CYP2C9* genotypes that lead to impaired *CYP2C9* function (**1/*2* and **2/*2*) was also detected to be higher in the cases than the controls (Table 5). Logistic regression analysis with hypoglycaemia status as the dependent variable and *CYP2C9* genotypes (**1/*2* and **2/*2*) as contributing variables estimated *CYP2C9*1/*2* and *CYP2C9*2/*2* genotype as risk factors for SIH. The confounding factors such as age, triglyceride levels, urea level, creatinine level, and mean daily dose of sulphonylurea medication were found to be uniform in the cases and controls. When adjusted for the given confounding factors, the association between the *CYP2C9*2* allele and its corresponding genotype with sulphonylurea-induced hypoglycaemia remained consistent, suggesting that the genetic variant may independently contribute to an increased risk of hypoglycaemia in individuals using sulphonylurea medications (for details, consult Table 5).

## 4. Discussion

Pakistan is among the most populous countries of the world, with a population of over 220 million people. The country has a rich cultural and ethnic heritage, with various ethnic groups (major ethnic groups include Pashtuns, Sindhi, Punjabi, and Balochi) contributing to its genetic diversity. Despite having one of the largest populations in the world, there is a scarcity of pharmacogenomic studies investigating genetic variations in different enzymes/genes that could alter drug response [19]. Cytochrome P450-2C9 (*CYP2C9*) is an important enzyme that is involved in the biotransformation and clearance of many drugs, including oral sulphonylureas. Different alleles of the *CYP2C9* gene have been identified, with some alleles associated with reduced enzyme activity and others associated with increased activity. Individuals carrying alleles associated with reduced *CYP2C9* activity may metabolize drugs, such as sulphonylureas, more slowly, leading to higher drug levels and an increased risk of adverse effects, including hypoglycaemia [20]. In the present study, we investigated the association of *CYP2C9*2* and **3* alleles and their genotypes with SIH in T2DM patients who were being treated with sulphonylurea drugs such as glimepiride or gliclazide. The findings of our study suggest that the presence of the *CYP2C9*2* allele (and its corresponding genotypes *CYP2C9*1/*2* and *CYP2C9*2/*2*) plays a key role in predisposing patients receiving sulphonylurea treatment toward hypoglycaemia. Furthermore, our study reported that the *CYP2C9*3* (loss-of-activity) allele was present at a ratio of 3.5% vs. 2.5% in cases vs. control. The frequency of the *CYP2C9*3* allele was found to be approximately uniform in cases vs. control. We also considered several additional factors that could potentially influence the distribution and clearance of sulphonylureas, thereby affecting the occurrence of hypoglycaemia. These factors included age, gender, cholesterol levels, creatinine level, urea level, and mean daily dose of glimepiride and gliclazide—the oral anti-diabetic medications. Importantly, the study found no significant differences in these parameters between the cases (patients who experienced hypoglycaemia) and the controls (patients who did not experience hypoglycaemia). This suggests that the observed association between the *CYP2C9*2* allele and hypoglycaemia is independent of these factors and is likely attributed to the genetic variation itself. Multiple independent studies have found consistent results and reported similar associations, which strengthens the evidence for the relationship between the *CYP2C9*2* allele and hypoglycaemia [21,22,23,24].

Studies investigating the effect of *CYP2C9* variants on the risk of sulphonylurea-related hypoglycaemia are relatively limited, and the results from the available studies have been inconsistent [25]. Some studies have reported an association between *CYP2C9* reduced-function alleles and an increased risk of hypoglycaemia in T2DM patients taking sulphonylureas. These findings suggest that individuals with specific *CYP2C9* genotypes may have a higher susceptibility to hypoglycaemia when treated with sulphonylurea drugs [26,27]. However, other studies have not detected any evidence of an association between *CYP2C9* variants and sulphonylurea-related hypoglycaemia [28,29]. This lack of consistency in findings may be due to various factors, including differences in study design, such as variations in the definition of hypoglycaemia, the age of the study population, the specific sulphonylureas included in the analysis, and the lack of statistical power resulting from small sample sizes.

Sulphonylurea-induced hypoglycaemia is a critical concern in the treatment of diabetes. It can lead to various complications, ranging from discomfort and reduced adherence to therapy to severe morbidity and mortality [30,31]. Minimizing the occurrence of hypoglycaemic episodes is essential for providing safe and effective treatment to diabetes patients. Genotyping for *CYP2C9* genetic variations can be particularly valuable, especially during the initiation of sulphonylurea treatment. Certain *CYP2C9* genotypes are associated with reduced enzyme activity, leading to slower metabolism of sulphonylureas and an increased risk of drug accumulation and subsequent hypoglycaemia. By identifying patients with *CYP2C9* genotypes that predict low enzyme activity, genotyping can aid in preventing medication overdose and subsequent hypoglycaemic episodes [32,33,34]. This information allows healthcare providers to adjust the dosage or choose alternative treatment options that are less likely to cause adverse effects in individuals with specific genotypes. Personalized medicine approaches, such as genotyping, can contribute to more precise and tailored therapies, minimizing the risk of adverse drug events and optimizing treatment outcomes for patients with diabetes.

## 5. Conclusions

In the present case–control perspective study, we explored the association of the *CYP2C9*2* allele (known to have low enzyme activity) with sulphonylurea-induced hypoglycaemia in the studied population. The *CYP2C9*1/*2* and *CYP2C9*2/*2* genotypes, as well as the *CYP2C9*2* allele, were found to be more prevalent in the cases (individuals experiencing SIH) compared to the control (individuals without such adverse reactions). It is important to note that further research and replication studies are needed to confirm and validate these findings. Additionally, the clinical implications of this association should be carefully considered in terms of personalized medicine, drug dosing, and patient management strategies for individuals with this reduced-function allele and its related genotypes.

## 6. Limitations

The study included participants from the Pashtun population only; inclusion of study participants from other Pakistani sub-populations such as Sindhi, Punjabi, and Balochi would have enhanced the diversity and generalizability of the study findings. We only focused on *CYP2C9* gene variants and left out other interesting Cytochrome P450 (*CYP450*) genes that affect many drugs metabolism and clearance. The sample size was limited to 400 individuals. Despite the mentioned limitations, the present study is the first of its kind in the Pakistani Pashtun population, and it identified important pharmacogenetic variants in the *CYP2C9* gene which are associated with sulphonylurea-induced hypoglycaemia.

## Figures and Tables

**Table 1 biomedicines-11-02282-t001:** Socio-demographic characteristics of cases and controls.

Variables	Cases *n* (f)	Control *n* (f)	*p*-Value
Gender			
Male	165 (82.5%)	173 (86.5%)	0.061
Female	35 (17.5%)	27 (13.5%)	
Mean age (y)	58 ± 12:40	56 ± 13:43	0.605
Mean weight (kg)	62.64 ± 6:07	59.55 ± 8:32	0.213
Address			0.318
Peshawar	45 (22.5%)	16 (16%)	
Charsadda	31 (15.5%)	13 (13%)	
Mardan	22 (11.0%)	13 (13%)	
Kohat	12 (6.0%)	11 (11%)	
Swabi	19 (9.5%)	4 (4%)	
Nowshera	17 (8.5%)	5 (5%)	
Bannu	18 (9.0%)	10 (%)	
karak	5 (2.5%)	2 (25%)	
Dir	16 (11%)	10 (2%)	
Swat	15 (7.5%)	10 (10%)	
Occupation			0.058
Business	30 (15.0%)	6 (6.0%)	
Govt. servant	37 (18.5%)	27 (27.0%)	
Retired	35 (17.5.0%)	30 (30.0%)	
Farming	25 (12.5%)	10 (10.0%)	
Housewife	40 (20.0%)	15 (15.0%)	
Labor	33 (16.5%)	12 (12.0%)	
Family Hx of T2DM			0.004
Yes	133 (66.5%)	175 (63%)	
No	67 (33.5%)	25 (25%)	
Marital status			0.138
Single	71 (35.5%)	43 (34%)	
Married	129 (64.5%)	157 (57%)	
Smoking			0.063
Yes	104 (52.0%)	80 (80%)	
No	96 (48%)	20 (20%)	
Naswar			0.061
Yes	130 (65.0)	153 (76.5%)	
No	70 (35.0%)	47 (23.5%)	
Diet and drug compliance			0.012
Yes	127 (63.5.5%)	42 (42%)	
No	73 (36.5)	58 (58%)	
Socioeconomic status			0.314
Good	52 (26.0%)	34 (34%)	
Average	102 (51.0%)	53 (53%)	
Below average	46 (23%)	13 (13%)	

*n* = number; f = frequency; kg = kilogram; y = years; Hx = history.

**Table 2 biomedicines-11-02282-t002:** Co-morbidities prevalence in study participants.

Co-Morbid Disease	Frequency (f)		*p*-Value
	Cases (*n* = 200)	Controls (*n* = 200)	
Hypertension	112.0%	102.5%	0.071
IHD	21.0%	18.0%	0.311
Renal Failure	5.0%	3.0%	0.412
Retinopathy	61.0%	58.91%	0.112
HBV	0.00%	0.00%	0.000
HCV	0.00%	0.00%	0.000

IHD: Ischemic heart disease; HCV: Hepatitis C virus; HBV: Hepatitis B virus.

**Table 3 biomedicines-11-02282-t003:** Bio-chemical characteristics of study participants.

Variables	Cases (*n* = 200)	Controls (*n* = 200)	*p*-Value
Total cholesterol (mg/dL)	265.25 ± 15.37	259.23 ± 12.54	0.076
LDL-cholesterol (mg/dL)	125.4 ± 14.12	121.62 ± 8.91	0.118
HDL-cholesterol (mg/dL)	58.2 ± 7.01	62.1 ± 5.22	0.132
Triglycerides (mg/dL)	158.3 ± 10.5	156.2 ± 9.15	0.184
Urea (mg/dL)	41.44 ± 15.20	39.50 ± 17.97	0.213
Creatinine (mg/dL)	0.94 ± 0.15	0.92 ± 0.21	0.982
HBA1C (%)	7.60 ± 5.55	7.39 ± 1.46	0.911
Mean daily dose of glimepiride (mg)	5.13 ± 1.56	5.13 ± 1.54	0.998
Mean daily dose of gliclazide (mg)	108.09 ± 44.37	105.00 ± 34.06	0.761

mg/dL: milligram per decilitre; mg: milligram; LDL: low density lipoprotein; HDL: high density lipoprotein.

**Table 4 biomedicines-11-02282-t004:** Allelic distribution/frequencies of CYPC19*2 and CYPC19*3 in cases and controls.

*CYP2C9* Alleles	Phenotypes	Cases *n* (f)	Controls *n* (f)	OR (95%Cl)	Crude *p*-Value	Adjusted *p*-Value
*CYP2C9*1*	Wild-type/no effect	158 (84.0%)	183 (91.5%)	Ref.	Ref.	Ref.
*CYP2C9*2*	Decreases enzyme activity	35 (17.5%)	12 (6.0%)	0.102 (0.08–3.08)	0.021	0.031
*CYP2C9*3*	Loss of enzyme activity	07 (3.5%)	5 (2.5%)	0.041 (0.02–2.21)	0.101	0.091

*n* = number; f = frequency.

**Table 5 biomedicines-11-02282-t005:** Genotype frequencies of CYPC19*2 and CYPC19*3 in cases and controls.

CYP2C9 Genotypes	Phenotypes	Type of Genotype	Cases *n* (f)	Controls *n* (f)	OR (95%Cl)	Crude *p*-Value	Adjusted *p*-Value
*CYP2C9 *1/*1*	Normal metabolizer	Hom	158(79%)	178 (89%)	Ref.	Ref.	Ref.
*CYP2C9 *1/*2*	Intermediate metabolizer	Het	20 (10%)	12 (6.0%)	1.21 (1.81–3.02)	0.036	0.039
*CYP2C9 *1/*3*	Intermediate metabolizer	Het	2 (1.0%)	3 (1.5%)	0.94 (0.77–1.15)	0.344	0.231
*CYP2C9 *2/*2*	Intermediate metabolizer	Hom	20 (7%)	7 (3.5%)	2.83 (1.69–3.00)	0.028	0.031
*CYP2C9 *2/*3*	Poor metabolizer	Het	---	---	---	---	--
*CYP2C9 *3/*3*	Poor metabolizer	Hom	---	---	---	---	---

Ref.: reference/wild-type; Hom: homozygous; Het: heterozygous.

## Data Availability

All necessary data/information are provided with the manuscript. However, for any additional data/information related to this article, the corresponding author can be contacted.

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
