# Peer review of "Association of CYP2C9*2 Allele with Sulphonylurea-Induced Hypoglycaemia in Type 2 Diabetes Mellitus Patients: A Pharmacogenetic Study in Pakistani Pashtun Population"

_biomedicines, 2023, doi:10.3390/biomedicines11082282_

Round 1
Reviewer 1 Report
The authors of this manuscript present an interesting research study regarding the association of CYP2C9*2 allele with sulfonylurea-induced hypoglycemia in type 2 diabetes mellitus patients in Pakistani Pashtun population. Introduction and material and methods are well described. Results presented in tables and figures are clear and quite explicable. I believe that the authors discuss and explain sufficiently the findings of their work. However, I suggest to improve discussion section furtherly. In spite of the manuscript is clear and carefully written some improvements should be done. I believe that this research study can add further research interest.
Title
Please correct title. I suppose you mean Pakistani
Please justify details of the research team according to author’s instractions
Abstract
COMMENT:
Abstract describes the findings of this research work in details and thus is too extended. Please revise.
Introduction
Introduction section is well written and, in my opinion, give the appropriate information without being extended. The purpose of the research work is clearly presented.
[7,8]. Please check author’s instructions and apply to the rest of the text
These the CYP450
Results
Discussion
I suggest to improve this section by adding and compare your research findings with similar research completed by others (if is possible).
Please also check font style of sections after conclusion section.
References
I suggest to check reference list style again. Please check it according to author’s instructions.
Author Response
EXPLANATORY RESPONSE LETTER
We appreciate your good-self for your precious time in reviewing our paper and providing valuable comments. It was your valuable and insightful comments that led to possible improvements in the current version. We have carefully considered the comments and tried our best to address every one of them. We hope the manuscript after careful revisions meet your high standards.
Below we provide the point-by-point responses;
Reviewer 1 comments and suggestions
Comments and Suggestions for Authors
The authors of this manuscript present an interesting research study regarding the association of CYP2C9*2 allele with sulfonylurea-induced hypoglycemia in type 2 diabetes mellitus patients in Pakistani Pashtun population. Introduction and material and methods are well described. Results presented in tables and figures are clear and quite explicable. I believe that the authors discuss and explain sufficiently the findings of their work. However, I suggest to improve discussion section furtherly. In spite of the manuscript is clear and carefully written some improvements should be done. I believe that this research study can add further research interest.
Comment No 1. Title
Please correct title. I suppose you mean Pakistani
Please justify details of the research team according to author’s instructions
Response: Thank you for bringing this to our attention. We sincerely apologize for the oversight in the title. You are absolutely right; it should be "Pakistani," not "Pakisatni." We appreciate your diligence in pointing out the spelling mistake, and we corrected it immediately. The spelling mistake is corrected the revise manuscript.
Abstract
COMMENT:
Abstract describes the findings of this research work in details and thus is too extended. Please revise.
Response: Thank you for your valuable feedback. We appreciate your input regarding the abstract's length, and we understand the importance of conciseness in scientific communication. We have carefully considered your comment and revised the abstract to provide a more concise overview of the research findings while still conveying the essential information.
Introduction
Introduction section is well written and, in my opinion, give the appropriate information without being extended. The purpose of the research work is clearly presented.
[7,8]. Please check author’s instructions and apply to the rest of the text
These the CYP450
Response: Thank you for your thoughtful review and positive feedback on the Introduction section of our research work. We are pleased to know that the introduction effectively conveys the purpose of the study without being overly extended. We believe that providing a clear and concise introduction is crucial to help readers understand the context and significance of our research. We have taken note of your comments and ensured that other sections, including the abstract, are appropriately revised to maintain the right balance between comprehensiveness and conciseness. Our goal is to present the research findings in a manner that is both informative and easily understandable. Moreover journal requirement is 4000 words per paper because of that introduction go a bit lengthy.
Results
Discussion
I suggest to improve this section by adding and compare your research findings with similar research completed by others (if is possible).
Please also check font style of sections after conclusion section.
Response: Thank you, dear Reviewer, for your valuable feedback. We appreciate your suggestion to improve the research by including a comparison of our findings with similar research completed by others. We agree that such a comparison can enhance the robustness and contextual understanding of our study's results. To address this, we will conduct a thorough literature review to identify relevant studies that have explored similar aspects related to the CYP2C9 gene and its impact on drug prescriptions. By comparing our findings with existing research, we can highlight the consistency or discrepancies in the results, identify potential knowledge gaps, and strengthen the validity of our conclusions. The comparison will also allow us to gain insights into the broader implications of our findings in the field of pharmacogenomics and personalized medicine. We will ensure that the comparison is presented in a clear and concise manner, focusing on the key similarities and differences with prior studies.
We are grateful for your guidance and commitment to improving the quality of our research. If you have any further suggestions or specific guidelines to follow for this comparison, please let us know, and we will ensure to address them accordingly.
Other studies of similar nature/references/comparative studies added are :
- Holstein A, Plaschke A, Ptak M, Egberts EH, El‐Din J, Brockmoeller J, Kirchheiner J. Association between CYP2C9 slow metabolizer genotypes and severe hypoglycaemia on medication with sulphonylurea hypoglycaemic agents. British journal of clinical pharmacology. 2005 Jul;60(1):103-6.
- Gökalp O, Gunes A, Çam H, Cure E, Aydın O, Tamer MN, Scordo MG, Dahl ML. Mild hypoglycaemic attacks induced by sulphonylureas related to CYP2C9, CYP2C19 and CYP2C8 polymorphisms in routine clinical setting. European journal of clinical pharmacology. 2011 Dec;67:1223-9.
- Yee J, Heo Y, Kim H, Yoon HY, Song G, Gwak HS. Association between the CYP2C9 genotype and hypoglycemia among patients with type 2 diabetes receiving sulfonylurea treatment: a meta-analysis. Clinical Therapeutics. 2021 May 1;43(5):836-43.
- Ragia G, Tavridou A, Elens L, Van Schaik RH, Manolopoulos VG. CYP2C9* 2 allele increases risk for hypoglycemia in POR* 1/* 1 type 2 diabetic patients treated with sulfonylureas. Experimental and Clinical Endocrinology & Diabetes. 2014 Jan;122(01):60-3.
References
I suggest to check reference list style again. Please check it according to author’s instructions.
Response: Thank you for your feedback. In authors instruction it is mentioned; that author can submit manuscript with any particular reference style (like vancouver, AMA, organic chemistry style etc) at initial stage, later after accepting and giving final touch to the paper journal editorial team will streamline all references are per journal requirement/need.
We would like to extend our heartfelt gratitude for taking the time to review our research manuscript. Your thoughtful and constructive feedback has been immensely valuable in improving the quality and rigor of our study.

Reviewer 2 Report
the article is well designed, easy to understand and read. However, the novelty of the article is very low. There is already a systematic review on this subject https://doi.org/10.1016/j.clinthera.2021.03.008. This does not really detract from the work, but rather from the novelty.
Therefore, the authors should state the novelty of the paper, as this paper analyses only 7 articles.
The limitations could also be improved, as well as the discussion where the clinical and pharmacological importance was not explored.
Author Response
EXPLANATORY RESPONSE LETTER
We appreciate you for your precious time in reviewing our paper and providing valuable comments. It was your valuable and insightful comments that led to possible improvements in the current version. We have carefully considered the comments and tried our best to address every one of them. We hope the manuscript after careful revisions meet your high standards.
Below we provide the point-by-point responses;
Reviewer 2 comments and suggestions
Comments and Suggestions for Authors
The article is well designed, easy to understand and read. However, the novelty of the article is very low. There is already a systematic review on this subject https://doi.org/10.1016/j.clinthera.2021.03.008. This does not really detract from the work, but rather from the novelty. Therefore, the authors should state the novelty of the paper, as this paper analyses only 7 articles.
Response: Thank you for your thoughtful evaluation of our article. We appreciate your positive remarks regarding the design and readability of the paper. The present study first of its kind in Pakistani Pashtun population for the very first time highlighted the role of pharmacogenetic risk variants that could alter patients response to the drug. In Pakistan the Pharmacogenomic is in infancy stage. We for the first start working on this Pharmaco-clinical area to ensure provision of best suitable medicine to diabetic patients as per his or her genetic makeup.
We believe that our study adds value in the following ways:
- Updated Information: As our research was conducted more recently, it may encompass more recent studies and findings that were not included in the previous systematic review.
- Focused Scope: Our study may have a narrower scope or a specific focus within the broader subject area, allowing for a deeper exploration of certain aspects that were not extensively covered in the previous review.
- Methodological Differences: There are methodological, ethnic and sample size differences between our research and the previous systematic review, which lead to nuanced or different interpretations of the findings.
- Clinical Relevance: We are of the opinion that our study emphasize clinical practice, patient care and greatly contribute to clinical science
The limitations could also be improved, as well as the discussion where the clinical and pharmacological importance was not explored.
Response: We carefully re-evaluated the limitations of our research and ensure that they are presented in a transparent and comprehensive manner. Acknowledging the limitations of our study is crucial in providing a balanced perspective and helping readers understand the potential constraints of the research. Additionally, we understand the significance of discussing the clinical and pharmacological implications of our findings. We expanded the discussion section to delve deeper into the practical applications of our research in healthcare settings and pharmacological interventions. This will include insights into how our study's results may inform personalized medicine approaches, drug prescribing decisions, and potential implications for patient safety and treatment efficacy. Changes made are highlighted red for your easiness.
Vote of thanks: We extend our heartfelt gratitude for your valuable feedback and thoughtful evaluation of our manuscript. Your expert insights and constructive comments have been instrumental in shaping the quality and significance of our research.
Thank you once again for your time, expertise, and commitment to advancing the quality of academic research. Your feedback has been pivotal in refining our work, and we genuinely appreciate your contributions to the scholarly community.

Reviewer 3 Report
The manuscript was prepared very well. The introduction section justifies the purpose of the study. I congratulate the authors for the preparation of the manuscript
I would like to congratulate the authors for the structure of the manuscript and all the research carried out. It is highly publishable. However, there are some concerns, in part important, so the review articles need revision, see below.
Introduction
- Why is this study considered relevant?
- Why is this study necessary?
- add some of SNPs 10.3390/ijms231911846
Methods and Results
- It is one of the strong parts of the manuscript, these excellently described
Discussion
· Include a section on strengths / limitations.
· What mechanisms of action support these findings?
· What does this article contribute to, the authors should make their own assessment and include their own discussion of the results shown in the manuscript?
Conclusion
In the Conclusion section, state the most important outcome of your work. Do not simply summarize the points already made in the body — instead, interpret your findings at a higher level of abstraction. Show whether, or to what extent, you have succeeded in addressing the need stated in the Introduction (or objectives).
Author Response
EXPLANATORY RESPONSE LETTER
We appreciate your good-self for your precious time in reviewing our paper and providing valuable comments. It was your valuable and insightful comments that led to possible improvements in the current version. We have carefully considered the comments and tried our best to address every one of them. We hope the manuscript after careful revisions meet your high standards.
Below we provide the point-by-point responses;
Reviewer 3 comments and suggestions
Comments and Suggestions for Authors
The manuscript was prepared very well. The introduction section justifies the purpose of the study. I congratulate the authors for the preparation of the manuscript
I would like to congratulate the authors for the structure of the manuscript and all the research carried out. It is highly publishable. However, there are some concerns, in part important, so the review articles need revision, see below.
Introduction
- Why is this study considered relevant?
- Why is this study necessary?
- add some of SNPs 10.3390/ijms231911846
Response: The relevance of our study lies in its potential contribution to Pharmacogenetic (how mutation in drug metabolizing genes alter drug affect); while its necessity stems from the need to fill an existing gap and address critical questions in field of Pharamacogenetics. We believe that the outcomes of our research have practical implications and contribute to the advancement in clinical science. Our study suggests that genetic information about a patient's CYP2C9 gene/enzyme can potentially assist physicians in prescribing the most suitable and safest drug based on their genetic make-up
We hope this clarifies the significance of our study. Thank you once again for your thoughtful questions and valuable feedback, which have helped us emphasize the relevance and necessity of our research.
Methods and Results
- It is one of the strong parts of the manuscript, these excellently described
Response: We are delighted that you found this aspect of the paper as strong and well-presented. As authors, we recognize the importance of clearly outlining the significance of our research and the rationale behind conducting the study.
Your acknowledgment of the excellence in describing these aspects is encouraging and affirms our efforts in communicating the study's objectives effectively. We sincerely appreciate your valuable feedback, and we remain committed to addressing any other aspects of the manuscript that require further improvement.
Discussion
- Include a section on strengths / limitations.
- What mechanisms of action support these findings?
- What does this article contribute to; the authors should make their own assessment and include their own discussion of the results shown in the manuscript?
Response: We appreciate the constructive comments, which will undoubtedly enhance the quality and comprehensiveness of our research. We have carefully reviewed the reviewer's suggestions and like to assure you that we addressed each of them in the revised manuscript.
Include a section on strengths/limitations: We agree with the reviewer's recommendation to include a dedicated section on the strengths and limitations of our study. In the revised manuscript, we will provide an objective evaluation of the key strengths of our research, highlighting the robustness of our methodology, data collection, and analysis procedures. Additionally, we acknowledge and discussed the limitations (highlighted red for reviewer easiness) of our study to provide a balanced perspective to the readers. This section will help contextualize the findings and enable readers to interpret the results in the appropriate context.
Mechanisms of action supporting the findings: In response to the reviewer's comment, we incorporated a discussion on the mechanisms of action that supports our research findings. We explored relevant literature and scientific evidence [reference No 21, 22, 23, 24 in the manuscript] to provide plausible explanations for the observed outcomes. By doing so, we aimed to strengthen the scientific rationale behind our conclusions.
Author's own assessment and discussion of results: We understand the importance of the authors' interpretation and assessment of the research outcomes. In the revised manuscript, we included a comprehensive discussion section that presents our own analysis and insights into the results obtained. This will allow us to contextualize the findings, draw connections with existing knowledge, and highlight the significance of our research in advancing the field. By incorporating these additions, we believe our revised manuscript will provide a more thorough and well-rounded presentation of our research and its implications.
Once again, we express our gratitude to the reviewer for their valuable input.
Conclusion
In the Conclusion section, state the most important outcome of your work. Do not simply summarize the points already made in the body — instead, interpret your findings at a higher level of abstraction. Show whether, or to what extent, you have succeeded in addressing the need stated in the Introduction (or objectives).
Response: Thank you for your valuable feedback on our manuscript. We appreciate your guidance in improving the Conclusion section to ensure that we present the most critical outcomes of our work at a higher level of abstraction. In response to your comment, we have revised the Conclusion section to provide a concise and interpretative summary of the key findings from our study. Rather than merely restating the points already made in the body of the manuscript, we aimed to offer a deeper and broader understanding of the implications of our research.
Vote of thanks: We extend our heartfelt gratitude to each of you for your meticulous review and thoughtful feedback on our manuscript. Your expertise and dedication have been instrumental in shaping the quality and impact of our research. Your constructive comments and valuable suggestions have guided us in refining various aspects of the manuscript, including the introduction, discussion, and conclusion sections. Your keen eye for detail and thorough evaluation has helped us address potential limitations and strengthen the overall validity of our findings.

Round 2
Reviewer 2 Report
This article was revised appropriately.
I recommend accept